# Structured Variational Inference in Continuous Cox Process Models

**Virginia Aglietti**
University of Warwick
The Alan Turing Institute
V.Aglietti@warwick.ac.uk

**Edwin V. Bonilla**
CSIRO's Data61
Edwin.Bonilla@data61.csiro.au

**Theodoros Damoulas**
University of Warwick
The Alan Turing Institute
T.Damoulas@warwick.ac.uk

**Sally Cripps**
Centre for Translational Data Science
The University of Sydney
Sally.Cripps@sydney.edu.au

## Abstract

We propose a scalable framework for inference in a continuous sigmoidal Cox process that assumes the corresponding intensity function is given by a Gaussian process (GP) prior transformed with a scaled logistic sigmoid function. We present a tractable representation of the likelihood through augmentation with a superposition of Poisson processes. This view enables a structured variational approximation capturing dependencies across variables in the model. Our framework avoids discretization of the domain, does not require accurate numerical integration over the input space and is not limited to GPs with squared exponential kernels. We evaluate our approach on synthetic and real-world data showing that its benefits are particularly pronounced on multivariate input settings where it overcomes the limitations of mean-field methods and sampling schemes. We provide the state of-the-art in terms of speed, accuracy and uncertainty quantification trade-offs.

## 1 Introduction

Point processes have been used effectively to model a variety of event data such as occurrences of diseases [9, 19], location of earthquakes [21] or crime events [2, 11] . The most commonly adopted class of models for such discrete data are non-homogenous Poisson processes and in particular Cox processes [6]. In these, the observed events are assumed to be generated from a Poisson point process (PPP) whose intensity is stochastic, enabling non-parametric inference and uncertainty quantification.

Gaussian processes [GPs; 25] form a flexible prior over functions and, therefore, have been used to model the intensity of a Cox process via a non-linear positive link function. Typical mappings are the exponential [9, 22], the square [17, 19] and the sigmoidal [1, 10, 12] transformations. In general, inferring the intensity function over a continuous input space $\mathcal{X}$ is highly problematic as it requires integrating an infinite-dimensional random function. This integral is generally intractable and, depending on the transformation used, different algorithms have been proposed to deal with this issue. For example, under the exponential transformation, a regular computational grid is commonly introduced [9]. While this significantly simplifies inference, it leads to poor approximations, especially in high dimensional settings. Increasing the resolution of the grid to improve the approximation yields computationally prohibitive algorithms that do not scale, highlighting the well-known trade-off between statistical performance and computational cost.

Other algorithms have been proposed to deal with a continuous $\mathcal{X}$ but they are computationally expensive [1, 12], are limited to simple covariance functions [19], require accurate numerical integra-

Table 1: Summary of related work. $\mathcal{X}$ represents the inputs space with $\int$ and $\sum$ denoting continuous and discrete models respectively. $\mathcal{O}$ gives the time complexity of the algorithm. $M$ represents the number of thinned points derived from the thinning [16] of a PPP. $K$ indicates the number of inducing inputs. STVB denotes our approach.

| | **STVB** | LGCP [22] | SGCP [1] | Gunter et al. [12] | VBPP [19] | Lian et al. [17] | MFVB [10] |
|---|---|---|---|---|---|---|---|
| **Inference** | SVI | MCMC | MCMC | MCMC | VI-MF | VI-MF | VI-MF |
| $\mathcal{O}$ | $K^3$ | $N^3$ | $(N+M)^3$ | $(N+M)^3$ | $NK^2$ | $NK^2$ | $NK^2$ |
| $\boldsymbol{\lambda(x)}$ | $\lambda^\star\sigma(f(x))$ | $\exp(f(x))$ | $\lambda^\star\sigma(f(x))$ | $\lambda^\star\sigma(f(x))$ | $(f(x))^2$ | $(f(x))^2$ | $\lambda^\star\sigma(f(x))$ |
| $\boldsymbol{\mathcal{X}}$ | $\int$ | $\sum$ | $\int$ | $\int$ | $\int$ | $\sum$ | $\int$ |

tion over the domain [10] or do not account for the model dependencies in the posterior distribution [10]. In this paper we propose an inference framework that addresses all of these modeling and inference limitations by having a tractable representation of the likelihood via augmentation with a superposition of PPPs. This enables a scalable structured variational inference algorithm (SVI) in the continuous space directly, where the approximate posterior distribution incorporates dependencies between the variables of interest. Our specific contributions are as follows.

**Scalable inference in continuous input spaces**: The augmentation of the input space via a process superposition view allows us to develop a scalable variational inference algorithm that does not require discretization or *accurate* numerical integration. With this view, we obtain a joint distribution that is readily normalized, providing a natural regularization over the latent variables in our model.

**Efficient structured posterior estimation**: We estimate a joint posterior that captures the complex variable dependencies in the model while being significantly faster than sampling approaches.

**State-of-the-art performance**: Our experimental evaluation shows the benefits of our approach when compared to state-of-the-art inference schemes, link functions, augmentation schemes and representations of the input space $\mathcal{X}$.

## 1.1 Related work

GP-modulated Poisson point processes are the gold standard for modeling event data. Performing inference in these models is challenging due to the need to integrate an infinite-dimensional random function over $\mathcal{X}$. Under the exponential transformation, inference has typically required discretization where the domain is gridded and the intensity function is assumed to be constant over each grid cell [4, 7, 9, 22]. Alternatively, Lasko [15] also considers an exponential link function and performs inference over a renewal process resorting to numerical integration within a computationally expensive sampling scheme. These methods suffer from poor scaling with the dimensionality of $\mathcal{X}$ and sensitivity to the choice of discretization or numerical integration technique. Several approaches have been proposed to deal with inference in the continuous domain directly by using alternative transformations along with additional modeling assumptions and computational tricks or by constraining the GP [20].

One of those alternative transformation is the squared mapping as developed in the Permanental process [13, 17–19, 28]. Although the square transformation enables analytical computation of the required integrals over $\mathcal{X}$, this only holds for certain standard types of kernels such as the squared exponential. In addition, Permanental processes suffer from important identifiability issues such as reflection invariance[1] and lead to models with "nodal lines" [13].

Another transformation is the scaled logistic sigmoid function proposed by [1] that achieves tractability by augmenting the input space via thinning [16], which can be seen as a point-process variant of rejection sampling. This model is known as the sigmoidal Gaussian Cox process (SGCP). Their proposed inference algorithm is based on Markov chain Monte Carlo (MCMC), which enables drawing 'exact' samples from the posterior intensity. However, as acknowledged by the authors, it has significant computational demands making it unfeasible to large datasets. As an extension to this work, [12] introduce the concept of "adaptive thinning" and propose an expensive MCMC scheme which scales as $\mathcal{O}(N^3)$. More recently, [10] introduced a neat double augmentation scheme for SGCP which enables closed form updates using a mean-field approximation (VI-MF). However, it

requires accurate numerical integration over $\mathcal{X}$, which makes the performance of the algorithm highly dependent on the number of integration points.

In this work, we overcome the limitations of the mentioned VI-MF and MCMC schemes by proposing an SVI framework, henceforth STVB, which takes into account the complex posterior dependencies while being scalable and thus applicable to high-dimensional real-world settings. To the best of our knowledge we are the first to propose a fast structured variational inference framework for GP modulated point process models. See Tab. 1 for a summary of the most relevant related works.

## 2 Model formulation

We consider learning problems where we are given a dataset of $N$ events $\mathcal{D} = \{\mathbf{x}_n\}_{n=1}^N$, where $\mathbf{x}_n$ is a $D$-dimensional vector in the compact space $\mathcal{X} \subset \mathbb{R}^D$. We aim at modeling these data via a PPP, inferring its underlying intensity function $\lambda(\mathbf{x}) : \mathcal{X} \to \mathbb{R}^+$ and making probabilistic predictions.

### 2.1 Sigmoidal Gaussian Cox process

Consider a realization $\xi = (N, \{\mathbf{x}_1, ..., \mathbf{x}_n\})$ of a PPP on $\mathcal{X}$ where the points $\{\mathbf{x}_1, ..., \mathbf{x}_n\}$ are treated as *indistinguishable* apart from their locations [8]. Conditioned on $\lambda(\mathbf{x})$, the Cox process likelihood function evaluated at $\xi$ can be written as:

$$\mathcal{L}(\xi|\lambda(\mathbf{x})) = \exp\left(-\int_{\mathcal{X}} \lambda(\mathbf{x})d\mathbf{x}\right) \prod_{n=1}^N \lambda(\mathbf{x}_n), \tag{1}$$

where the intensity is given by $\lambda(\mathbf{x}) = \lambda^\star \sigma(f(\mathbf{x}))$ with $\lambda^\star > 0$ being an upperbound on $\lambda(\mathbf{x})$ with prior distribution $p(\lambda^\star)$, $\sigma(\cdot)$ denoting the the logistic sigmoid function and $f$ is drawn from a zero-mean GP prior with covariance function $\kappa(\mathbf{x}, \mathbf{x}'; \boldsymbol{\theta})$ and hyperparameters $\boldsymbol{\theta}$, i.e. $f|\boldsymbol{\theta} \sim \mathcal{GP}(\mathbf{0}, \kappa(\mathbf{x}, \mathbf{x}'; \boldsymbol{\theta}))$. We will refer to this joint model as the sigmoidal Gaussian Cox process (SGCP). Notice that, when considering the tuple $(\mathbf{x}_1, ..., \mathbf{x}_n)$ instead of the set $\{\mathbf{x}_1, ..., \mathbf{x}_n\}$, and thus the event $\xi_0 = (N, (\mathbf{x}_1, ..., \mathbf{x}_n))$, the likelihood function is given by $\mathcal{L}(\xi_0|\lambda(\mathbf{x})) = \frac{\mathcal{L}(\xi|\lambda(\mathbf{x}))}{N!}$. There are indeed $N!$ permutations of the events $\{\mathbf{x}_1, ..., \mathbf{x}_n\}$ giving the same point process realization. When the set $\{\mathbf{x}_1, ..., \mathbf{x}_n\}$ is known, considering $\mathcal{L}(\xi|\lambda(\mathbf{x}))$ or $\mathcal{L}(\xi_0|\lambda(\mathbf{x}))$ does not affect the inference procedure. The same holds for MCMC algorithms inferring the event locations. In this case, the factorial term disappears in the computation of the acceptance ratio. However, as we shall see later, when the event locations are latent variables in a model and inference proceeds via a variational approximation the difference between the two likelihoods is essential. Indeed, while $\mathcal{L}(\xi_0|\lambda(\mathbf{x}))$ is normalized with respect to $N$, one must be cautious when integrating the likelihood in Eq. (1) over sets and bring back the missing $N!$ factor so as to obtain a proper discrete probability mass function for $N$.

As it turns out, inference in SGCP is *doubly intractable*, as it requires solving the integral in Eq. (1) and computing the intractable posterior distribution for the latent function at the $N$ event locations and the bounding intensity, i.e. $p(\mathbf{f}_N, \lambda^\star|\{\mathbf{x}_n\}_{n=1}^N)$, which in turns requires computing the marginal likelihood. One way to avoid the first source on intractability (integral in Eq. (1)) is through augmentation of the input space [1, 10], a procedure that introduces precisely those latent (event) variables that require explicit normalization during variational inference. We will describe below a process superposition view of this augmented scheme that allows us to define a proper distribution over the joint space of observed and latent variables and carry out posterior estimation via variational inference. By superimposing two PPP with opposite intensities we obtain an homogenous PPP and thus avoid the integration of the GP over $\mathcal{X}$ while reducing the integral in Eq. (1) to the computation of the measure of the input space $\int_{\mathcal{X}} d\mathbf{x}$.

### 2.2 Augmentation via superposition

A very useful property of independent PPPs is that their superposition, which is defined as the combination of events from two processes in a single one, is a PPP. Consider two PPP with intensities $\lambda(\mathbf{x})$ and $\nu(\mathbf{x})$ and realisations $(N, \{\mathbf{x}_1, ..., \mathbf{x}_n\})$ and $(M, \{\mathbf{y}_1, ..., \mathbf{y}_M\})$ respectively. The combined event $\xi_R = (R = M + N, \{\mathbf{v}_1, ..., \mathbf{v}_R\})$ is a realization of a PPP with intensity given by $\lambda(\mathbf{x}) + \nu(\mathbf{x})$ where knowledge of which points originated from which process is assumed lost. The likelihood for

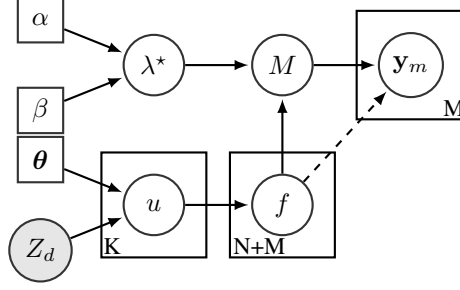

Figure 1: Plate diagram representing the posterior distribution accounting for all model dependencies. In our variational posterior (Eq. (6)) we drop the dependency represented by the dashed line.

$\mathcal{L}(\xi_R | \lambda(\mathbf{x}), \nu(\mathbf{x}))$ can be thus written as:

$$\sum_{N=0}^{R} \binom{N+M}{N} \sum_{P_N \in \mathbb{P}_N} \left( \frac{\exp(-\int_{\mathcal{X}} \lambda(\mathbf{x})d\mathbf{x})}{N!} \prod_{r \in P_N} \lambda(r) \times \frac{\exp(-\int_{\mathcal{X}} \nu(\mathbf{x})d\mathbf{x})}{M!} \prod_{r \in P_N^c} \nu(r) \right),$$

(2)

where $\mathbb{P}_N$ denotes the collection of all possible partitions of size $N$, $P_N$ represents an element of $\mathbb{P}_N$ and $P_N^c$ is its complement.

Consider now $R = N + M$ to be the total number of events resulting from thinning [16] where $N$ is the number of observed events while $M$ is the number of latent events with stochastic locations $\mathbf{y}_1, ..., \mathbf{y}_M$. We assume that the probability of observing an event is given by $\sigma(f(\mathbf{x}))$ while the probability for the event to be latent is $\sigma(-f(\mathbf{x}))$. In addition, let $\lambda^\star \int_{\mathcal{X}} d\mathbf{x}$ be the expected total number of events. We can see the realization $(M + N, (\mathbf{x}_1, ..., \mathbf{x}_N, \mathbf{y}_1, ..., \mathbf{y}_M))$ as the result of the superposition of two PPPs with intensities $\lambda(\mathbf{x}) = \lambda^\star \sigma(f(\mathbf{x}))$ and $\nu(\mathbf{x}) = \lambda^\star \sigma(-f(\mathbf{x}))$. Differently from the standard superposition, we do know which events are observed and which are latent. In writing the likelihood for $(M + N, \{\mathbf{x}_1, ..., \mathbf{x}_N, \mathbf{y}_1, ..., \mathbf{y}_M\})$ we thus do not need to consider all the possible partitions of $N$. We can write $\mathcal{L}_{N+M} \overset{\text{def}}{=} \mathcal{L}(N + M, (\mathbf{x}_1, ..., \mathbf{x}_N, \mathbf{y}_1, ..., \mathbf{y}_M))$:

$$\mathcal{L}_{N+M} = \frac{\exp(-\int_{\mathcal{X}} \lambda(\mathbf{x})d\mathbf{x})}{N!} \prod_{r \in P_N} \lambda(r) \times \frac{\exp(-\int_{\mathcal{X}} \nu(\mathbf{x})d\mathbf{x})}{M!} \prod_{r \in P_N^c} \nu(r)$$

(3)

$$= \frac{1}{N!M!} \exp(-\lambda^\star \int_{\mathcal{X}} d\mathbf{x})(\lambda^\star)^{M+N} \prod_{n=1}^{N} \sigma(f(\mathbf{x}_n)) \prod_{m=1}^{M} \sigma(-f(\mathbf{y}_m)).$$

(4)

The augmentation via superposition offers a different view on the thinning procedure proposed in Adams et al. [1]. However, there is a crucial difference between Eq. 4 and the usual likelihood considered in SGCP [1]. Eq. (4) represents a distribution over tuples and thus, as mentioned above, is properly normalized. In addition, it makes a distinction between the observed and latent events and it is thus different from Eq. (1) written for the the tuple $(M + N, \{\mathbf{x}_1, ..., \mathbf{x}_N, \mathbf{y}_1, ..., \mathbf{y}_M\})$. We can write the full joint distribution as $\mathcal{L}_{N+M}^{+} \overset{\text{def}}{=} \mathcal{L}(\{\mathbf{x}_n\}_{n=1}^N, \{\mathbf{y}_m\}_{m=1}^M, M, \mathbf{f}, \lambda^\star | \mathcal{X}, \boldsymbol{\theta})$:

$$\mathcal{L}_{N+M}^{+} = \frac{(\lambda^\star)^{N+M} \exp(-\lambda^\star \int_{\mathcal{X}} d\mathbf{x})}{N!M!} \prod_{n=1}^{N} \sigma(f(\mathbf{x}_n)) \prod_{m=1}^{M} \sigma(-f(\mathbf{y}_m)) \times p(\mathbf{f}) \times p(\lambda^\star),$$

(5)

where $p(\mathbf{f}) \overset{\text{def}}{=} p(\mathbf{f}_{N+M})$ denotes the joint prior at both $\{\mathbf{x}_n\}_{n=1}^N$ and $\{\mathbf{y}_m\}_{m=1}^M$ and $p(\lambda^\star)$ denotes the prior over the upper bound of $\lambda(\mathbf{x})$. We consider $p(\lambda^\star) = \text{Gamma}(a, b)$ and set $a$ and $b$ so that $\lambda^\star$ has mean and standard deviation equal to $2\times$ and $1\times$ the intensity we would expect from an homogenous PPP on $\mathcal{X}$. Eq. (5) represents the joint distribution for the data and the variables in the model. Estimating their posterior distributions requires computing the marginal likelihood by integrating out all variables in Eq. (5). This is generally intractable and in section §3 we perform inference via a variational approximation which minimises a bound, the so-called evidence lower bound (ELBO), to the marginal likelihood.

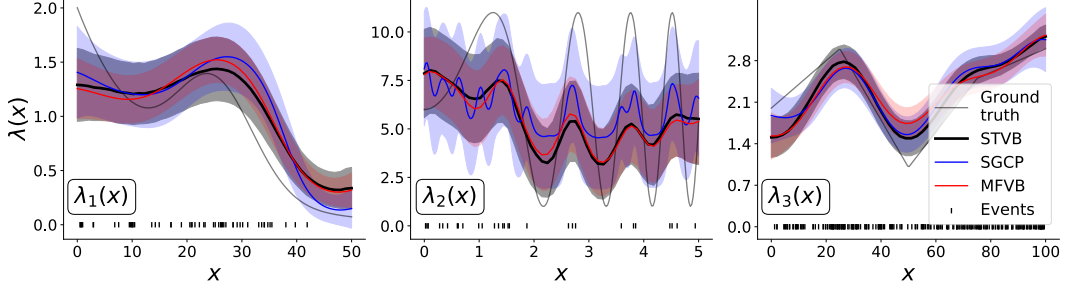

Figure 2: Qualitative results on synthetic data. Solid colored lines denote posterior mean intensities while shaded areas are $\pm$ standard deviation.

## 2.3 Scalability via inducing variables

As in standard GP modulated models, the introduction of a GP prior poses significant computational challenges during posterior estimation as inference is dominated by algebraic operations that are cubic on the number of observations. In order to make inference scalable, we follow the inducing-variable approach proposed by [27] and further developed by [3]. To this end, we consider an augmented prior $p(\mathbf{f}, \mathbf{u})$ with $K$ underlying inducing variables denoted by $\mathbf{u}$. The corresponding inducing inputs are given by the $K \times D$ matrix $\mathbf{Z}$. Major computational gains are realized when $K \ll N + M$. The augmented prior distributions for the inducing variables and the latent functions are $p(\mathbf{u}|\boldsymbol{\theta}) = \mathcal{N}(\mathbf{0}, \mathbf{K_{ZZ}})$ and $p(\mathbf{f}|\mathbf{u}, \boldsymbol{\theta}) = \mathcal{N}(\mathbf{K_{XZ}}(\mathbf{K_{ZZ}})^{-1}\mathbf{u}, \mathbf{K_{XX}} - \mathbf{A}\mathbf{K_{ZX}})$ where $\mathbf{A} = \mathbf{K_{XZ}}(\mathbf{K_{ZZ}})^{-1}$ and $\mathbf{X}$ denotes the $(N + M) \times D$ matrix of all events locations $\{\mathbf{x}_n, \mathbf{y}_m\}_{n=1,m=1}^{N,M}$. $\mathbf{K_{UV}}$ is the covariance matrix obtained by evaluating the covariance function at all pairwise columns of matrices $\mathbf{U}$ and $\mathbf{V}$.

## 3 Structured Variational Inference in the augmented space

Given the joint distribution in Eq. 5, our goal is to estimate the posterior distribution over all latent variables given the data. i.e. $p(\mathbf{f}, \mathbf{u}, M, \{\mathbf{y}_m\}_{m=1}^{M}, \lambda^\star | \mathcal{D})$. This posterior is analytically intractable and we resort to variational inference [14]. Variational inference entails defining an approximate posterior $q(\mathbf{f}, \mathbf{u}, M, \{\mathbf{y}_m\}_{m=1}^{M}, \lambda^\star)$ and optimizing the ELBO with respect to this distribution. In SGCP, the GP and the latent variables are highly coupled and breaking their dependencies would lead to poor approximations, especially in high dimensional settings. Fig. 1 shows the structure of a general posterior distribution for SGCP without any factorisation assumption. We consider an approximate posterior distribution that takes dependencies into account:

$$Q(\mathbf{f}, \mathbf{u}, M, \{\mathbf{y}_m\}_{m=1}^{M}, \lambda^\star) = p(\mathbf{f}|\mathbf{u})q(\{\mathbf{y}_m\}_{m=1}^{M}|M)q(M|\mathbf{f}, \lambda^\star)q(\mathbf{u})q(\lambda^\star) \qquad (6)$$

With respect to the general posterior distribution, the only factorisation we impose in Eq. (6) is in the factor $q(\{\mathbf{y}_m\}_{m=1}^{M}|M)$ where we drop the dependency on $\mathbf{f}$, see dashed line in Fig. 1. We set:

$$q(\mathbf{u}) = \mathcal{N}(\mathbf{m}, \mathbf{S}) \qquad q(\lambda^\star) = \text{Gamma}(\alpha, \beta) \qquad q(\{\mathbf{y}_m\}_{m=1}^{M}|M) = \prod_{m=1}^{M} \sum_{s=1}^{S} \pi_s \mathcal{N}_T(\mu_s, \sigma_s^2; \mathcal{X})$$

where $\mathcal{N}_T(\cdot; \mathcal{X})$ denotes a truncated Gaussian distribution on $\mathcal{X}$. The factorisation assumption between $\mathbf{f}$ and $\{\mathbf{y}_m\}_{m=1}^{M}$ can be relaxed by considering a PPP with intensity $\lambda^\star\sigma(-f(\mathbf{x}))$ as the joint variational distribution $q(M, \{\mathbf{y}_m\}_{m=1}^{M})$, which is indeed the true posterior distributions for the number of latent events and their locations [8]. Considering a fully structured posterior distribution significantly increases the computational cost of the algorithm as it would require sampling from the full posterior in the computation of the ELBO. The mixture of truncated Gaussians provides a flexible and computationally advantageous alternative while satisfying the constraint of being within the domain of interest.

More importantly, we assume $q(M|\mathbf{f}, \lambda^\star) = \text{Poisson}(\eta)$ with $\eta = \lambda^\star \int_{\mathcal{X}} \sigma(-f(\mathbf{x}))d\mathbf{x}$. This is indeed the *true* conditional posterior distribution for the number of latent points, see Proposition (3.7) in [23]. Considering $q(M|\mathbf{f}, \lambda^\star)$ we thus fully account for the dependency structure existing among

$M$, $\mathbf{f}$ and $\lambda^\star$. Crucially, while in this work we estimate $\int_\mathcal{X} \sigma(-f(\mathbf{x}))d\mathbf{x}$ via Monte Carlo, STVB does not require *accurate* estimation of this term. Indeed, differently from the competing techniques [10], where the algorithm convergence and the posterior $q(\mathbf{f})$ is *directly* dependent on numerical integration, STVB only requires evaluation of the integral during optimisation but $q(\mathbf{f})$ and thus $\lambda(\mathbf{x})$ do not directly depend on its value. In other words, the quality of the posterior intensity does not depend directly on the accurate estimation of this integral.

## 3.1 Evidence Lower Bound

Following standard variational inference arguments, it is straightforward to show that the ELBO decomposes as:

$$\mathcal{L}_{\text{elbo}} = N(\psi(\alpha) - \log(\beta)) - V\frac{\alpha}{\beta} - \log(N!) + \underbrace{\mathbb{E}_Q[M\log(\lambda^\star)]}_{T_1} - \underbrace{\mathbb{E}_Q[\log(M!)]}_{T_2} +$$

$$+ \sum_{n=1}^{N} \mathbb{E}_{q(\mathbf{f})}[\log(\sigma(f(\mathbf{x}_n)))] + \underbrace{\mathbb{E}_Q\left[\sum_{m=1}^{M}\log(\sigma(-f(\mathbf{y}_m)))\right]}_{T_3} - \mathcal{L}_{\text{kl}}^{\mathbf{u}} - \mathcal{L}_{\text{kl}}^{\lambda^\star} - \underbrace{\mathcal{L}_{\text{ent}}^M}_{T_4} - \underbrace{\mathcal{L}_{\text{ent}}^{\{\mathbf{y}_m\}_{m=1}^M}}_{T_5}$$

$$(7)$$

where $V = \int_\mathcal{X} d\mathbf{x}$, $\psi(\cdot)$ is the digamma function and $q(\mathbf{f}) = \mathcal{N}(\mathbf{Am}, \mathbf{K_{XX}} - \mathbf{AK_{ZX}} + \mathbf{ASA}^T)$. The terms denoted by $T_i, i = 1, ..., 5$ cannot be computed analytically. Naïvely, black-box variational inference algorithms could be used to estimate these terms via Monte Carlo, thus sampling from the full variational posterior in Eq. (6). This would require sampling $\mathbf{f}$, $\lambda^\star$, $M$ and $\{\mathbf{y}_m\}_{m=1}^M$ thus slowing down the algorithm while leading to slow convergence. On the contrary, we exploit the structure of the model and the approximate posterior to simplify these terms and increase the algorithm efficiency. Denote $\mu(\mathbf{f}) = \int_\mathcal{X} \sigma(-f(\mathbf{x}))d\mathbf{x}$, we can write:

$$T_1 = \mathbb{E}_{q(\lambda^\star)}[\lambda^\star \log(\lambda^\star)]\mathbb{E}_{q(\mathbf{f})}[\mu(\mathbf{f})], \quad T_5 = \frac{\alpha}{\beta}\mathbb{E}_{q(\mathbf{y}_m)}[\log q(\mathbf{y}_m)]\mathbb{E}_{q(\mathbf{f})}[\mu(\mathbf{f})] \tag{8}$$

$$T_3 = \frac{\alpha}{\beta}\mathbb{E}_{q(\mathbf{f})}[\mu(\mathbf{f})]\mathbb{E}_{q(\mathbf{f})q(\mathbf{y}_m)}[\log(\sigma(-f(\mathbf{y}_m)))], \tag{9}$$

$$T_4 = \frac{\alpha}{\beta}\mathbb{E}_{q(\mathbf{f})}[\mu(\mathbf{f})[\log(\mu(\mathbf{f})) - 1]] + \mathbb{E}_{q(\lambda^\star)}[\lambda^\star\log(\lambda^\star)]\mathbb{E}_{q(\mathbf{f})}[\mu(\mathbf{f})] - \mathbb{E}_Q[\log(M!)]. \tag{10}$$

Notice how the term $-\mathbb{E}_Q[\log(M!)]$ in $T_4$, which would require further approximations, appears with opposite sign in $T_2$ (Eq. (7)) and thus cancels out in the computation of the ELBO. See the supplement (§1) for the full derivations.

Eqs. (8)–(10) give an expression for $\mathcal{L}_{\text{elbo}}$ which avoids sampling from $q(M|\mathbf{f}, \lambda^\star)$ and $q(\{\mathbf{y}_m\}_{m=1}^M|M)$ and does not require computing the GP on the stochastic locations of the latent events. The remaining expectations are with respect to reparameterizable distributions. We thus avoid the use of other estimators (such as the score function estimators) which would lead to high-variance gradient estimates. Stochastic optimisation techniques can be used to evaluate $T_3$ and $\sum_{n=1}^N \mathbb{E}_{q(\mathbf{f})}[\log(\sigma(f(\mathbf{x}_n)))]$ in Eq. (7) thus reducing the computational cost by making it independent of $M$ and $N$. This would reduce the computational complexity of the algorithm to $\mathcal{O}(K^3)$. However, when the number of inputs used per mini-batch equals $N$, the time complexity becomes $\mathcal{O}(NK^2)$. In the following experiments, we show how the proposed structured approach together with these efficient ELBO computations lead to higher predictive performances and better uncertainty quantification. The presented results do not exploit the computational gains attainable via stochastic optimisation thus the CPU times and performances are directly comparable across all methods.

## 4 Experiments

We test our algorithm on three 1D synthetic data settings and on two 2D real-world applications[2].

**Baselines** We compare against alternative inference schemes, different link functions and a different augmentation scheme. In terms of continuous models, we consider a sampling approach [SGCP, 1], a Permanental Point process model [VBPP, 19] and a mean-field approximation based on a Pólya-Gamma augmentation [MFVB, 10]. In addition, we compare against a discrete variational log-Gaussian Cox process model [LGCP, 24]. Details are given in the supplement (§3).

**Performance measures** We test the algorithms evaluating the $l_2$ norm to the true intensity function (for the synthetic datasets), the test log likelihood ($\ell_{test}$) on the test set and the negative log predicted likelihood (NLPL) on the training set. In order to assess the model capabilities in terms of uncertainty quantification, we compute the empirical coverage (EC), i.e. the coverage of the empirical count distributions obtained by sampling from the posterior intensity. We do that for different credible intervals (CI) on both the training (in-sample, $p(N|\mathcal{D})$) and test set (out-of-sample, $p(N^*|\mathcal{D})$). Details on the metrics are in the supplement (§2). For the synthetic data experiments, we run the algorithms with 10 training datasets each including a different PPP realization sampled from the ground truth. For each different training set, we then evaluate the performance on other 10 unseen realizations sampled again from the ground truth. We compute the mean and the standard deviation for the presented metrics averaging across the training and test sets. For the real data settings, we compute the NLPL and in-sample EC on the observed events. We then test the algorithm computing both $\ell_{test}$ and out-of-sample EC on the held-out events. In order to compute the out-of-sample EC we rescale the intensity function as $\lambda_{test}(\mathbf{x}) = \lambda_{train}(\mathbf{x}) - N_{train}/V + N_{test}/V$ with $V = \int_{\mathcal{X}} d\mathbf{x}$. We then sample from $\lambda_{test}(\mathbf{x})$ and generate the predicted count distributions for different seeds.

**Synthetic experiments** We test our approach using the three toy example proposed by [1]:

- $\lambda_1(x) = 2\exp(-1/15) + \exp(-[(x-15)/10]^2)$   $x \in [0, 50]$,
- $\lambda_2(x) = 5\sin(x^2) + 6$   $x \in [0, 5]$ and
- $\lambda_3(x)$ piecewise linear through $(0, 20)$, $(25, 3)$, $(50, 1)$, $(75, 2.5)$, $(100, 3)$   $x \in [0, 100]$.

For LGCP, we discretize the input space considering a grid cell width of one for $\lambda_1(x)$ and $\lambda_3(x)$ and of 0.5 for $\lambda_2(x)$. For MFVB we consider 1000 integration points. In terms of $q(\{\mathbf{y}_m\}_{m=1}^M|M)$, we set $S = 5$ but consistent results where found across different values of this parameter. The results are given in Fig. 2 and Tab. 2, where we see that all algorithms recover similar predicted mean intensities and give roughly comparable performances across all metrics. Out of all 9 settings and metrics (top section of Tab. 2) our method (STVB) outperforms competing methods on 3 cases and it is only second to SGCP on 6 cases. However, the CPU time of SGCP is almost an order of magnitude larger than ours even in these simple low-dimensional problems. This confirms the benefits of having structured approximate posteriors within a computationally efficient inference algorithm such as VI. In terms of uncertainty quantification (bottom section of Tab. 2), our algorithm outperforms all competing approaches for $\lambda_1(x)$ and $\lambda_2(x)$.

**2D real data experiments** In this section we show the performance of the algorithm on two 2D real-world datasets. In both cases, we assume independent two-dimensional truncated Gaussian distributions for $q(\{\mathbf{y}_m\}_{m=1}^M|M)$ so that they factorize across input dimensions. Qualitative and quantitative results are given in Fig. 3, Fig. 4 and Tab. 3.

Our first dataset is concerned with neuronal data, where event locations correspond to the position of a mouse moving in an arena when a recorded cell fired [5, 26]. We randomly assign the events to either training ($N = 583$) or test ($N = 29710$) and we run the model using a regular grid of $10 \times 10$ inducing inputs. We see that the intensity function recovered by the three methods vary in terms of smoothness with MFVB estimating the smoothest $\lambda(\mathbf{x})$ and VBPP recovering an irregular surface (Fig. 3). MFVB gives slightly better performance in terms of $\ell_{test}$ but our method (STVB) outperforms competing approaches in terms of NLPL and EC figures. Remarkably, STVB contains the true number of test events in the 30% credible intervals for 56% of the simulations from the posterior intensity (Tab. 3 and Fig. 4).

As a second dataset, we consider the Porto taxi dataset[3] which contains the trajectories of 7000 taxi travels in the years 2013/2014 in the city of Porto. As in [10], we consider the pick-up locations as observations of a PPP and restrict the analysis to events happening within the coordinates $(41.147, -8.58)$ and $(41.18, -8.65)$. We select $N = 1000$ events at random as training set and train

Table 2: Average performances on synthetic data across 10 training and 10 test datasets with standard errors in brackets. Top: Lower values of $l_2$, NLPL and higher values of $\ell_{test}$ are better. Bottom: Out-of-sample EC for different CI, higher values are better. Our method denoted by STVB.

|  | $\lambda_1(x)$ | | | $\lambda_2(x)$ | | | $\lambda_3(x)$ | | | CPU time (s) |
|---|---|---|---|---|---|---|---|---|---|---|
|  | $l_2$ | $\ell_{test}$ | NLPL | $l_2$ | $\ell_{test}$ | NLPL | $l_2$ | $\ell_{test}$ | NLPL |  |
| STVB | **3.44** (1.43) | **-1.39** (1.05) | 4.71 (0.51) | 46.28 (9.95) | 56.04 (4.47) | 5.62 (0.72) | **7.39** (2.76) | 153.98 (11.91) | 6.41 (0.64) | 315.59 |
| MFVB | 4.56 (1.43) | -2.84 (1.0) | 4.74 (0.1) | 44.44 (10.7) | 55.35 (4.72) | 5.52 (1.29) | 8.17 (3.43) | 155.08 (10.20) | 5.82 (0.61) | 0.01 |
| VBPP | 9.19 (2.32) | -7.71 (3.31) | 8.91 (1.19) | 48.15 (13.16) | **56.82** (4.42) | 5.20 (1.33) | 20.54 (6.53) | 152.82 (11.43) | 8.35 (2.28) | 0.44 |
| SGCP | 4.22 (1.88) | **-1.39** (1.28) | **4.21** (1.04) | **43.50** (8.69) | 55.05 (1.35) | **3.77** (0.54) | 14.44 (2.97) | **165.66** (2.12) | **4.78** (0.33) | 2764.88 |
| LGCP | 67.76 (24.38) | -5.26 (8.84) | 26.26 (8.09) | 106.74 (13.89) | 28.56 (6.88) | 15.75 (3.36) | 19.24 (6.44) | 147.67 (11.76) | 10.84 (1.36) | 4.74 |

|  | EC$-\lambda_1(x)$ | | | EC$-\lambda_2(x)$ | | | EC$-\lambda_3(x)$ | | |
|---|---|---|---|---|---|---|---|---|---|
|  | 30% CI | 40% CI | 50% CI | 30% CI | 40% CI | 50% CI | 30% CI | 40% CI | 50% CI |
| STVB | **0.81** (0.27) | **0.72** (0.27) | **0.6** (0.34) | **0.91** (0.24) | **0.88** (0.23) | **0.86** (0.22) | **0.99** (0.03) | 0.97 (0.09) | 0.92 (0.15) |
| MFVB | 0.76 (0.25) | 0.61 (0.28) | 0.52 (0.29) | 0.89 (0.23) | 0.84 (0.29) | 0.82 (0.29) | 0.97 (0.09) | 0.91 (0.14) | 0.78 (0.15) |
| VBPP | 0.75 (0.21) | 0.41 (0.25) | 0.04 (0.09) | 0.76 (0.26) | 0.45 (0.26) | 0.05 (0.05) | 0.83 (0.19) | 0.43 (0.14) | 0.03 (0.05) |
| SGCP | 0.39 (0.28) | 0.27 (0.22) | 0.08 (0.12) | 0.64 (0.09) | 0.14 (0.05) | 0.00 (0.00) | 0.49 (0.03) | 0.34 (0.07) | 0.02 (0.04) |
| LGCP | 0.08 (0.12) | 0.03 (0.09) | 0.01 (0.03) | 0.04 (0.08) | 0.00 (0.00) | 0.00 (0.00) | **0.99** (0.00) | **0.99** (0.12) | **0.95** (0.10) |

Table 3: Average performances on real-data experiments with standard errors in brackets. EC is computed across 100 replications using different seeds. Higher $\ell_{test}$, EC and lower NLPL are better. EC figures are given as In-sample - Out-of-sample.

|  | Neuronal data | | | | | Taxi data | | | | |
|---|---|---|---|---|---|---|---|---|---|---|
|  | $\ell_{test}[\times 10^3]$ | NLPL | EC-30% CI | EC-40% CI | CPU time (s) | $\ell_{test}[\times 10^6]$ | NLPL $[\times 10^4]$ | EC-30% CI | EC-40% CI | CPU time (s) |
| STVB | -84.55 (16.05) | **10.10** (7.02) | **1.00-1.00** (0.00)-(0.00) | **0.99-0.56** (0.10)-(0.50) | 193.07 | **-27.96** (9.16) | **27.96** (9.16) | 0.81-**0.37** (0.39)-(0.48) | 0.09-**0.01** (0.29)-(0.10) | 290.34 |
| MFVB | **-83.54** (4.60) | 10.71 (3.39) | **1.00**-0.03 (0.00)-(0.17) | 0.78-0.00 (0.41)-(0.00) | 0.35 | -40.8 (6.41) | 40.65 (6.41) | 0.00-0.00 (0.00)-(0.00) | 0.00-0.00 (0.00)-(0.00) | 0.24 |
| VBPP | -83.89 (12.49) | 11.39 (8.18) | **1.00**-0.00 (0.00) - (0.00) | 0.83-0.00 (0.38)-(0.00) | 26.23 | -31.32 (8.18) | 31.32 (8.18) | **0.98**-0.00 (0.14)-(0.00) | **0.48**-0.00 (0.50)-(0.00) | 3.62 |

the model with 400 inducing points placed on a regular grid. The test log likelihood is then computed on the remaining 3401 events. We see that our method (STVB) outperforms competing methods on all performance metrics (Tab. 3), recovering an intensity that is smoother than VBPP and captures more structure compared to MFVB (Fig. 3). In terms of uncertainty quantification, the coverage of $p(N^*|\mathcal{D})$ are the highest for STVB across all CI. Notice how the irregularity of the VBPP intensity leads to good performance on the training set but results in a $p(N^*|\mathcal{D})$ which is centered on a significantly higher number of test events (Fig. 4). As expected, the SVI approach implies wider counts distributions compared to the mean field approximation. This generally yields better predictive performances in a variety of settings and especially in higher dimensions.

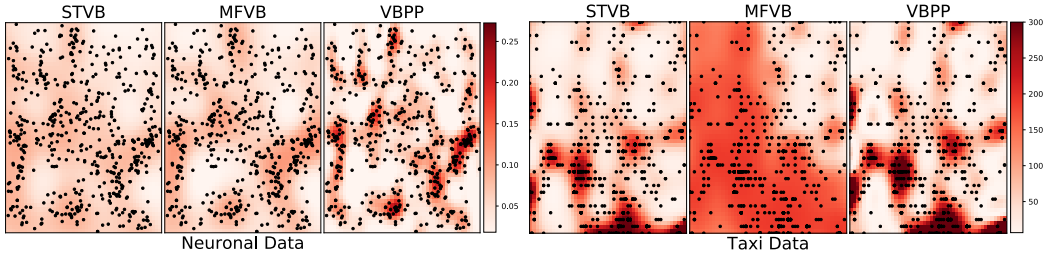

Figure 3: Real data. Posterior mean intensities and events on the two-dimensional input space.

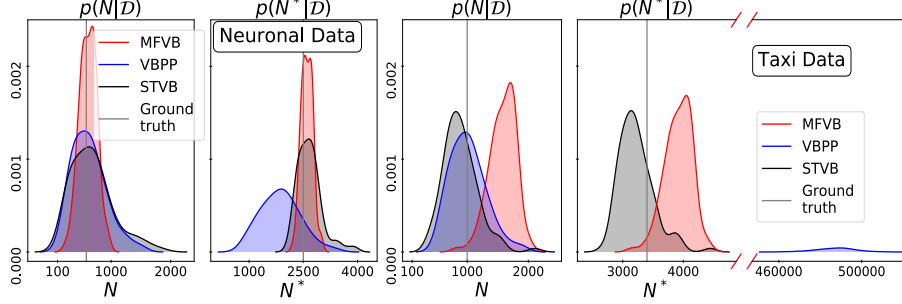

Figure 4: Predicted counts distributions for the training set ($p(N|\mathcal{D})$) and the test set ($p(N^*|\mathcal{D})$) on real data. The gray line denotes the number of observed events. The red bars on the x-axis denote breaks in the axis due to the different shifts of the distributions.

Table 4: Average performances on the spatio-temporal Taxi dataset. Standard errors in brackets. EC is computed across 100 replications using different seeds. Higher $\ell_{test}$, EC and lower NLPL are better. EC figures are given as In-sample - Out-of-sample.

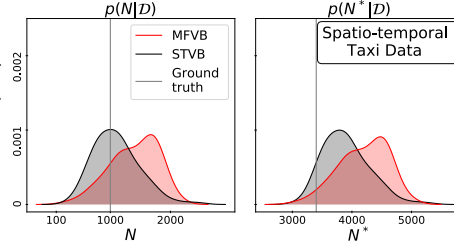

Figure 5: Predicted counts distributions for the training set ($p(N|\mathcal{D})$) and the test set ($p(N^*|\mathcal{D})$).

| | Spatio-temporal Taxi Data | | | | |
|---|---|---|---|---|---|
| | $\ell_{test}[\times 10^7]$ | NLPL$[\times 10^5]$ | EC-30% CI | EC-40% CI | CPU time (s) |
| STVB | **-31.26** (10.88) | **31.26** (10.88) | **1.00**-0.00 (0.00)-(0.00) | **0.98**-0.00 (0.14)-(0.00) | 1208.00 |
| VBPP | -42.97 (9.56) | 42.97 (9.56) | 0.00-0.00 (0.00)-(0.00) | 0.00-0.00 (0.00)-(0.00) | 1.00 |

**3D real data experiment** Finally, we show the performance of the algorithm on the spatio-temporal Taxi dataset used above where, for each taxi travel, we consider both the trajectory and the pickup time in seconds. While VBPP does not currently support D > 2, we found STVB to outperform MFVB both in terms of performance metrics and uncertainty quantification (Tab. 4, Fig. 5).

## 5 Conclusions and discussion

We have proposed a new variational inference framework for estimating the intensity of a continuous sigmoidal Cox process. By seeing an augmented input space from a superposition of two PPPs, we have derived a scalable and computationally efficient structured variational approximation. Our framework does not require discretization or accurate numerical computation of integrals on the input space, it is not limited to specific kernel functions and properly accounts for the strong dependencies existing across the latent variables. Through extensive empirical evaluation we have shown that our methods compares favorably against 'exact' but computationally costly MCMC schemes, while being almost an order of magnitude faster. More importantly, our inference scheme outperforms all competing approaches in terms of uncertainty quantification. The benefit of the proposed scheme and resulting SVI are particularity pronounced on multivariate input settings where accounting for the highly coupled variables become crucial for interpolation and prediction. Future work will focus on relaxing the factorization assumption between the GP and the latent points in the posterior. Introducing a fully structured variational inference would further improve the accuracy performance of the method but would require further approximations in the variational objective.

**Acknowledgments**

This work was supported by the EPSRC grant EP/L016710/1, The Alan Turing Institute under EPSRC grant EP/N510129/1, the Lloyds Register Foundation programme on Data Centric Engineering, the University of Sydney's Centre for Translational Data Science and the Australian Research Council ARC FT140101266.

## Footnotes

[1]With reflection invariance we refer to the invariance of the intensity function with respect to the sign change of the GP used to model it.

[2]Code and data for all the experiments is provided at `https://github.com/VirgiAgl/STVB`.

[3] http://www.geolink.pt/ecmlpkdd2015-challenge/dataset.html.

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
