[Supplementary Material]

# Supplementary material for Structured Variational Inference in Continuous Cox Process Models

**Virginia Aglietti**
University of Warwick
The Alan Turing Institute
V.Aglietti@warwick.ac.uk

**Edwin V. Bonilla**
CSIRO's Data61
Edwin.Bonilla@data61.csiro.au

**Theodoros Damoulas**
University of Warwick
The Alan Turing Institute
T.Damoulas@warwick.ac.uk

**Sally Cripps**
Centre for Translational Data Science
The University of Sydney
Sally.Cripps@sydney.edu.au

## 1 ELBO derivations

Here we derive the expressions given in Eqs. (7)-(10). As given in Eq. (7) the evidence lower bound ($\mathcal{L}_{\text{elbo}}$) decomposes as:

$$
\begin{aligned}
\mathcal{L}_{\text{elbo}} &= \mathbb{E}_Q\left[\log\left[\frac{p(\{\mathbf{x}_n\}_{n=1}^N, \{\mathbf{x}_m\}_{m=1}^M, M, \mathbf{f}, \mathbf{u}, \lambda^\star|\tau, \boldsymbol{\theta})}{p(\mathbf{f}|\mathbf{u})q(\{\mathbf{x}_m\}_{m=1}^M|M)q(M|\mathbf{f}, \lambda^\star)q(\mathbf{u})q(\lambda^\star)}\right]\right] \\
&= \mathbb{E}_Q\left[\log p(\{\mathbf{x}_n\}_{n=1}^N, \{\mathbf{x}_m\}_{m=1}^M, M, \mathbf{u}, \lambda^\star|\tau, \boldsymbol{\theta})\right] - \mathbb{E}_Q\left[\log q(\{\mathbf{x}_m\}_{m=1}^M|M)q(M|\mathbf{f}, \lambda^\star)q(\mathbf{u})q(\lambda^\star)\right] \\
&= \mathbb{E}_Q\left[(N+M)\log(\lambda^\star) - \lambda^\star\mu(\tau) - \log(M!) - \log(N!) + \sum_{n=1}^N \log(\sigma(f(\mathbf{x}_n)))\right] \\
&\quad + \mathbb{E}_Q\left[\sum_{m=1}^M \log(\sigma(-f(\mathbf{x}_m))) + \log(p(\mathbf{u})) + \log(p(\lambda^\star))\right] \\
&\quad - \mathbb{E}_Q\left[\log(q(\mathbf{u})) - \log(q(M|\mathbf{f}, \lambda^\star)) - \log(q(\lambda^\star)) - \log(q(\{\mathbf{x}_m\}_{m=1}^M|M))\right] \\
&= N(\psi(\alpha) - \log(\beta)) - V\frac{\alpha}{\beta} - \log(\log N!) + \underbrace{\mathbb{E}_Q[M\log(\lambda^\star)]}_{T_1} - \underbrace{\mathbb{E}_Q[\log M!]}_{T_2} + \\
&\quad + \sum_{n=1}^N \mathbb{E}_{q(\mathbf{u})}[\log(\sigma(f(\mathbf{x}_n)))] + \underbrace{\mathbb{E}_Q\left[\sum_{m=1}^M \log(\sigma(-f(\mathbf{x}_m)))\right]}_{T_3} + \\
&\quad - KL(q(\mathbf{u})||p(\mathbf{u})) - KL(q(\lambda^\star)||p(\lambda^\star)) \\
&\quad - \underbrace{\mathbb{E}_Q[\log q(M|\mathbf{f}, \lambda^\star)]}_{T_4} - \underbrace{\mathbb{E}_Q\left[\log q(\{x_m\}_{m=1}^M|M)\right]}_{T_5}
\end{aligned}
$$

Let's now focus on the terms $T_i, i = 1, ..., 5$.

The term $T_1$ (Eq. (8)) is given by:

$$\begin{aligned} T_1 &= \mathbb{E}_{q(\mathbf{u})q(\lambda^\star)}\big[\mathbb{E}_{q(M|\mathbf{u},\lambda^\star)}[M\log(\lambda^\star)]\big] \\ &= \mathbb{E}_{q(\mathbf{u})q(\lambda^\star)}\big[\log(\lambda^\star)\mathbb{E}_{q(M|\mathbf{u},\lambda^\star)}[M]\big] \\ &= \mathbb{E}_{q(\mathbf{u})q(\lambda^\star)}\left[\log(\lambda^\star)\lambda^\star\int_\mathcal{X}\sigma(-f(\mathbf{x}))d\mathbf{x}\right] \\ &= \mathbb{E}_{q(\lambda^\star)}[\lambda^\star\log(\lambda^\star)]\mathbb{E}_{q(\mathbf{f})}[\mu(\mathbf{f})] \end{aligned}$$

The term $T_3$ (Eq. (9)) is given by:

$$\begin{aligned} T_3 &= \mathbb{E}_{q(\mathbf{f})q(\mathbf{y}_m)q(\lambda^\star)}\left[\mathbb{E}_{q(M|\mathbf{f},\mathbf{y}_m,\lambda^\star)}\left[\sum_{m=1}^{M}\log(\sigma(-f(\mathbf{y}_m)))\right]\right] \\ &= \mathbb{E}_{q(\mathbf{f})q(\mathbf{y}_m)q(\lambda^\star)}\left[\log(\sigma(-f(\mathbf{y}_m)))\mathbb{E}_{q(M|\mathbf{f},\lambda^\star)}\left[\sum_{m=1}^{M}1\right]\right] \\ &= \mathbb{E}_{q(\mathbf{f})q(\mathbf{y}_m)q(\lambda^\star)}[\log(\sigma(-f(\mathbf{y}_m)))\lambda^\star\mu(\mathbf{f})] \\ &= \frac{\alpha}{\beta}\mathbb{E}_{q(\mathbf{f})}[\mu(\mathbf{f})]\mathbb{E}_{q(\mathbf{f})q(\mathbf{y}_m)}[\log(\sigma(-f(\mathbf{y}_m)))] \end{aligned}$$

The term $T_4$ (Eq. (10)) is given by:

$$\begin{aligned} T_4 &= \mathbb{E}_Q[-\lambda^\star\mu(\mathbf{f})] + \mathbb{E}_Q[M\log(\lambda^\star\mu(\mathbf{f}))] - \mathbb{E}_Q[\log M!] \\ &= -\frac{\alpha}{\beta}\mathbb{E}_{q(\mathbf{f})}[\mu(\mathbf{f})] + \mathbb{E}_{q(\lambda^\star)q(\mathbf{f})}[\lambda^\star\log(\lambda^\star)\mu(\mathbf{f}) + \lambda^\star\mu(\mathbf{f})\log(\mu(\mathbf{f}))] - \mathbb{E}_Q[\log M!] \\ &= -\frac{\alpha}{\beta}\mathbb{E}_{q(\mathbf{f})}[\mu(\mathbf{f})] + \mathbb{E}_{q(\lambda^\star)}[\lambda^\star\log(\lambda^\star)]\mathbb{E}_{q(\mathbf{f})}[\mu(\mathbf{f})] + \frac{\alpha}{\beta}\mathbb{E}_{q(\mathbf{f})}[\mu(\mathbf{f})\log(\mu(\mathbf{f}))] - \mathbb{E}_Q[\log(M!)] \end{aligned}$$

Finally, the term $T_5$ (Eq. (9)) is given by:

$$\begin{aligned} T_5 &= \mathbb{E}_Q\left[\sum_{m=1}^{M}\log q(\mathbf{y}_m)\right] \\ &= \mathbb{E}_{q(\mathbf{f},\lambda^\star,M)}\left[\sum_{m=1}^{M}\mathbb{E}_{q(\mathbf{y}_m)}[\log q(\mathbf{y}_m)]\right] \\ &= \mathbb{E}_{q(\mathbf{f},\lambda^\star,M)}[M]\mathbb{E}_{q(\mathbf{y}_m)}[\log q(\mathbf{y}_m)] \\ &= \mathbb{E}_{q(\mathbf{f})q(\lambda^\star)}[\lambda^\star\mu(\mathbf{f})]\mathbb{E}_{q(\mathbf{y}_m)}[\log q(\mathbf{y}_m)] \\ &= \frac{\alpha}{\beta}\mathbb{E}_{q(\mathbf{y}_m)}[\log q(\mathbf{y}_m)]\mathbb{E}_{q(\mathbf{f})}[\mu(\mathbf{f})] \end{aligned}$$

Notice how the last term in $T_4$ that is $-\mathbb{E}_Q[\log(M!)]$, appears with opposite sign in $T_2 = \mathbb{E}_Q[\log(M!)]$. This term is thus cancelling out in the computation of the ELBO.

## 2 Performance metrics

We test the algorithms evaluating the $l_2$ norm to the true intensity function (in the synthetic settings), the test log likelihood ($\ell_{test}$) on the test set and the negative log predicted likelihood (NLPL) on the training set. These metrics are computed as follow:

$$l_2 = \int_\mathcal{X}(\lambda(\mathbf{x}) - \bar{\lambda}(\mathbf{x}))^2 d\mathbf{x}$$

where $\lambda(\mathbf{x})$ is the true intensity function, $\bar{\lambda}(\mathbf{x})$ is the posterior mean intensity and the integral is evaluated numerically.

Table 1: Synthetic data $\lambda_1(\mathbf{x})$ - EC performance on training and test dataset. Higher values are better.

| | $\lambda_1(\mathbf{x})$ - In-sample EC | | | | | $\lambda_1(\mathbf{x})$ - Out-of-sample EC | | | | |
|---|---|---|---|---|---|---|---|---|---|---|
| | 10% CI | 20% CI | 30% CI | 40% CI | 50% CI | 10% CI | 20% CI | 30% CI | 40% CI | 50% CI |
| STVB | **1.00** | **1.00** | **1.00** | **1.00** | **1.00** | 0.96 | 0.88 | **0.81** | **0.72** | **0.60** |
| | (0.00) | (0.00) | (0.00) | (0.00) | (0.00) | (0.24) | (0.24) | (0.23) | (0.29) | (0.29) |
| MFVB | **1.00** | **1.00** | **1.00** | **1.00** | **1.00** | 0.95 | 0.80 | 0.76 | 0.61 | 0.52 |
| | (0.00) | (0.00) | (0.00) | (0.00) | (0.00) | (0.00) | (0.00) | (0.00) | (0.00) | (0.00) |
| VBPP | **1.00** | **1.00** | **1.00** | **1.00** | 0.10 | **1.00** | **0.97** | 0.75 | 0.41 | 0.04 |
| | (0.00) | (0.00) | (0.00) | (0.00) | (0.30) | (0.00) | (0.05) | (0.21) | (0.25) | (0.09) |
| SGCP | **1.00** | **1.00** | **1.00** | **1.00** | 0.60 | 0.75 | 0.60 | 0.39 | 0.27 | 0.08 |
| | (0.00) | (0.00) | (0.00) | (0.00) | (0.49) | (0.29) | (0.33) | (0.28) | (0.22) | (0.12) |
| LGCP | 0.70 | 0.00 | 0.00 | 0.00 | 0.00 | 0.48 | 0.22 | 0.08 | 0.03 | 0.01 |
| | (0.46) | (0.00) | (0.00) | (0.00) | (0.00) | (0.00) | (0.00) | (0.00) | (0.00) | (0.00) |

$$\ell_{test} = \mathbb{E}_{q(\lambda^\star)q(\mathbf{f})}\left[\log\left[\exp\left(-\int_{\mathcal{X}}\lambda(\mathbf{x})d\mathbf{x}\right)\prod_{\mathbf{x}\in\mathcal{D}_{\text{test}}}\lambda(\mathbf{x})\right]\right]$$

where again the integral is computed via numerical integration.

The NLPL is computed as

$$\text{NLPL} = -\frac{1}{S}\sum_{s=1}^{S}\log p(N_{\text{train}}|\int_{\mathcal{X}}\lambda^s(\mathbf{x})d\mathbf{x})$$

where $S$ denotes the number of samples from the variational distributions $q(\mathbf{f})$ and $q(\lambda^\star)$.

Finally, the EC is computed by evaluating the coverage of the CIs of the posterior ($p(N|\mathcal{D})$) and predictive ($p(N^*|\mathcal{D})$). To construct the empirical count distribution we sample from the variational distributions $q(\mathbf{f})$ and $q(\lambda^\star)$, obtain samples of $\lambda(\mathbf{x})$ and simulate $N$ or $N^*$ from $\text{Poisson}(\lambda^\star\int_{\mathcal{X}}\sigma(f(\mathbf{x}))d\mathbf{x})$.

## 3 Additional experimental results

For all comparisons we consider a GP with squared-exponential covariance function with equally set hyperparameters. Denote by $\boldsymbol{\theta}_i = (l, \sigma_f^2)$ the values of the hyperameters for the kernel function $K(\mathbf{x}, \mathbf{x}')$ on $\lambda_i(\mathbf{x})$ where $l$ indicates the lenghtscale. We set:

- $\boldsymbol{\theta}_1 = (10, 1)$
- $\boldsymbol{\theta}_2 = (0.25, 1)$
- $\boldsymbol{\theta}_3 = (15, 1)$

For the real-world settings we have:

- $\boldsymbol{\theta}_{\text{neuronal data}} = (10, 1)$
- $\boldsymbol{\theta}_{\text{taxi data}} = (0.3, 1)$
- $\boldsymbol{\theta}_{\text{spatio-temporal taxi data}} = (0.3, 1)$

### 3.1 Synthetic data experiments

In Tab. 1, 2 and 3 we report the values of EC for different CIs and on both the training and test set.

### 3.2 Real data experiments

In Tab. 4 we report the values of EC for different CIs and on both the training and test set.

Table 2: Synthetic data $\lambda_2(\mathbf{x})$ - EC performance on training and test dataset. Higher values are better.

| | $\lambda_2(\mathbf{x})$ - In-sample EC | | | | | $\lambda_2(\mathbf{x})$ - Out-of-sample EC | | | | |
|---|---|---|---|---|---|---|---|---|---|---|
| | 10% CI | 20% CI | 30% CI | 40% CI | 50% CI | 10% CI | 20% CI | 30% CI | 40% CI | 50% CI |
| STVB | **1.00** | **1.00** | **1.00** | **1.00** | **1.00** | **1.00** | **0.97** | **0.91** | **0.88** | **0.86** |
| | (0.00) | (0.00) | (0.00) | (0.00) | (0.00) | (0.00) | (0.09) | (0.24) | (0.23) | (0.22) |
| MFVB | **1.00** | **1.00** | **1.00** | **1.00** | **1.00** | 0.92 | 0.92 | 0.89 | 0.84 | 0.82 |
| | (0.00) | (0.00) | (0.00) | (0.00) | (0.00) | (0.00) | (0.00) | (0.00) | (0.00) | (0.00) |
| VBPP | **1.00** | **1.00** | **1.00** | **1.00** | 0.10 | 0.92 | 0.86 | 0.76 | 0.45 | 0.05 |
| | (0.00) | (0.00) | (0.00) | (0.00) | (0.30) | (0.24) | (0.23) | (0.26) | (0.26) | (0.05) |
| SGCP | **1.00** | 0.90 | 0.70 | 0.40 | 0.30 | 0.90 | 0.90 | 0.64 | 0.14 | 0.00 |
| | (0.00) | (0.30) | (0.46) | (0.49) | (0.46) | (0.00) | (0.00) | (0.09) | (0.05) | (0.00) |
| LGCP | 0.10 | 0.00 | 0.00 | 0.00 | 0.00 | 0.80 | 0.22 | 0.04 | 0.00 | 0.00 |
| | (0.30) | (0.00) | (0.00) | (0.00) | (0.00) | (0.24) | (0.16) | (0.08) | (0.00) | (0.00) |

Table 3: Synthetic data $\lambda_3(\mathbf{x})$ - EC performance on training and test dataset. Higher values are better.

| | $\lambda_3(\mathbf{x})$ - In-sample EC | | | | | $\lambda_3(\mathbf{x})$ - Out-of-sample EC | | | | |
|---|---|---|---|---|---|---|---|---|---|---|
| | 10% CI | 20% CI | 30% CI | 40% CI | 50% CI | 10% CI | 20% CI | 30% CI | 40% CI | 50% CI |
| STVB | **1.00** | **1.00** | **1.00** | **1.00** | **1.00** | **1.00** | **1.00** | 0.99 | 0.97 | 0.92 |
| | (0.00) | (0.00) | (0.00) | (0.00) | (0.00) | (0.00) | (0.00) | (0.00) | (0.00) | (0.12) |
| MFVB | **1.00** | **1.00** | **1.00** | **1.00** | **1.00** | **1.00** | **1.00** | 0.97 | 0.91 | 0.78 |
| | (0.00) | (0.00) | (0.00) | (0.00) | (0.00) | (0.00) | (0.00) | (0.00) | (0.00) | (0.00) |
| VBPP | **1.00** | **1.00** | **1.00** | **1.00** | 0.10 | 0.97 | 0.94 | 0.83 | 0.43 | 0.03 |
| | (0.00) | (0.00) | (0.00) | (0.00) | (0.30) | (0.09) | (0.15) | (0.19) | (0.14) | (0.05) |
| SGCP | 0.80 | 0.70 | 0.50 | 0.40 | 0.00 | 0.82 | 0.54 | 0.49 | 0.34 | 0.02 |
| | (0.40) | (0.46) | (0.50) | (0.49) | (0.00) | (0.12) | (0.05) | (0.03) | (0.07) | (0.04) |
| LGCP | **1.00** | **1.00** | **1.00** | **1.00** | **1.00** | **1.00** | **1.00** | **1.00** | **1.00** | **0.95** |
| | (0.00) | (0.00) | (0.00) | (0.00) | (0.00) | (0.00) | (0.00) | (0.00) | (0.00) | (0.00) |

Table 4: Real data. Values are given as In-sample - Out-of-sample EC. Mean and standard errors (in parenthesis) are computed across different seeds.

| | Neuronal data | | | | |
|---|---|---|---|---|---|
| | 10% EC | 20% EC | 30% EC | 40% EC | 50%EC |
| STVB | **1.00-1.00** | **1.00-1.00** | **1.00-1.00** | **0.99-0.56** | **0.01**- 0.00 |
| | 0.00 - 0.00 | 0.00 - 0.00 | 0.00-0.00 | (0.10)-(0.50) | (0.10)-0.00 |
| MFVB | **1.00-1.00** | **1.00**-0.62 | **1.00**-0.03 | 0.78-0.00 | 0.00 - 0.00 |
| | 0.00 - 0.00 | 0.00-(0.49) | 0.00-(0.17) | (0.41)-0.00 | 0.00 - 0.00 |
| VBPP | **1.00**-0.53 | **1.00**-0.00 | **1.00**-0.00 | 0.83-0.00 | **0.01**-0.00 |
| | 0.00-(0.50) | 0.00 - 0.00 | 0.00 - 0.00 | (0.38)-0.00 | (0.10)-0.00 |
| | Taxi data | | | | |
| | 10% EC | 20%EC | 30% EC | 40% EC | 50% EC |
| STVB | **1.00**-1.00 | **1.00-1.00** | 0.81-**0.37** | 0.09-**0.01** | 0.00-0.00 |
| | 0.00-0.00 | 0.00-0.00 | (0.39)-(0.48) | (0.29)-(0.10) | 0.00-0.00 |
| MFVB | 0.49-0.93 | 0.00-0.13 | 0.00-0.00 | 0.00-0.00 | 0.00-0.00 |
| | (0.50)-(0.26) | 0.00-(0.34) | 0.00-0.00 | 0.00-0.00 | 0.00-0.00 |
| VBPP | **1.00**-0.00 | **1.00**-0.00 | **0.98**-0.00 | **0.48**-0.00 | 0.00- 0.00 |
| | 0.00- (0.00) | 0.00-(0.00) | (0.14)-(0.00) | (0.50)-0.00 | 0.00-0.00 |