[Reviews · NeurIPS 2019]

Reviewer 1



In general, this paper is well-written and has important contributions to the machine learning community. Though it does not relieve the factorization assumption (line 260), it indeed provides a potential research direction in the future. I believe this paper should be at least among the top 50% of the accepted NeurIPS papers and would highly recommend this paper for acceptance.

Reviewer 2



This paper is attempts to tackle the integral in eq. (1) with augmentation. This is an original and signifiance contribution. The paper can be improved by adding/clarifying the following points 1. In section 2.2. It will make the paper clearer if the paper can state exactly the marginal likelihood to optimised via the ELBO. 2. Line 155: The approximate posterior of $y$ is a mixture of truncated Gaussian. This seems to be a model imposed on $y$, rather than an optimal form that comes from the variational optimisation and factorization. It'll be good to at least show what is optimal structure for the the model on $y$ and how close this structure is to the mixture of truncated Gaussian. 3. Line 157: Please clarify exactly what the integral of the sigmoid of the *vector* function $u$ at $x$ is? Is this also not one manner of discretization of the domain (that the abstract claims to avoid)? 4. Eq. 8: The inequality for $T_2$ is only approximate, since the Stirling's approximation is used. Please prove the exact inequality; otherwise, you'd have to relaxed the ELBO (i.e., lower bound) claim. 5. Section 4. I'll rather that the paper focus on the augmentation and the variational inference. This will make it easier to compare the CPU time among the different methods in the experiments. The paper can point to the possibility of using stochastic optimisation and delegate the associated results to the supplementary. 6. Figure 4. STVB is missing or mislabeled in this figure. 7. The brackets in multiple equations in the supplementary material are mismatched/missing.

Reviewer 3



The manuscript is well written and the presented approach is sufficiently original. The superposition of two independent PPs to remove the double intractability is new, but the variational inference algorithm follows from standard arguments.

[Author Response · NeurIPS 2019]

We thank the reviewers for their helpful feedback and suggestions that will significantly improve the final version of the paper. The reviewers agree that the paper is well-written (R1, R3) and offers an original and significant contribution to the machine learning community (R1, R2, R3). In particular, the paper proposes an efficient framework for Cox processes that overcomes the limitations of the current state-of-the-art schemes (R1). The proposed technique is free from the curse of dimensionality, preserves most of the dependencies in the model and does not rely on assumptions on the covariance function (R3). Furthermore, the algorithm is scalable and offers competitive experimental results (R2).

We hope our detailed response below will further highlight the paper's quality and originality and persuade them to increase their overall scores. For this, we have attached the tag *Q5* to some of our responses, indicating that we address Question 5: Improvements suggested by the reviewers that may yield to a score increase.

**R#1:** *(1) Factorization between the latent process and the latent locations*. We can relax this assumption by considering a PPP with intensity $\int_{\mathcal{X}} \lambda^{\star} \sigma(-f(\mathbf{x})) d\mathbf{x}$ as the joint variational distribution $q^{*}(M, \{\mathbf{y}_m\}_{m=1}^{M})$, which is indeed the true posterior [8]. In fact, we have implemented this approach (post-submission) and found out that while fully capturing model dependencies, it introduces a significant computational burden due to sampling from the full approximate posterior in the computation of $T_3$. Empirically, this full posterior yields better uncertainty quantification but comparable point-performance metrics to those reported in the paper. Increasing the computational efficiency of this new approach remains an interesting research direction.

(2) *Q5: Higher dimensions*. We thank reviewer for suggesting testing our algorithm on higher-dimensional data. While VBPP [19] does not currently support $D > 2$, we run our algorithm on the spatio-temporal Taxi dataset and found it to outperform MFVB [10] both in terms of performance metrics and uncertainty quantification, e.g. $\ell_{test}[\times 10^7] : -31.26$ vs $-42.97$. We will add this comparison in the additional page of the final version.

**R#2:** *(1) Clarity on marginal likelihood being optimized*. This corresponds to integrating out all latent variables in Eq. 5 (after including the augmented GP prior), which is analytically intractable. However, we will show its relationship to the ELBO explicitly in the final version. Many thanks for the suggestion.

(2) *Q5: Optimal structure of $q(\mathbf{y})$*. As mentioned above, the optimal joint distribution $q^{*}(M, \{\mathbf{y}_m\}_{m=1}^{M})$ is a PPP with intensity $\int_{\mathcal{X}} \lambda^{\star} \sigma(-f(\mathbf{x})) d\mathbf{x}$ , which we found to have comparable point-performance metrics. Critically, this fully structured posterior significantly increases the computational cost. The mixture of truncated Gaussians provides a flexible and computationally advantageous alternative, while satisfying the constraint of being within the domain of interest. See R#1 (1) above for more details.

(3) *Integral in line 157*: We estimate this using Monte Carlo. As described in lines 159–163, the key to our approach, which distinguishes it from previous work, is that this integral does not need to be estimated accurately, as we only require it during optimization and, therefore, the quality of the posterior intensity does not depend directly on how accurate this estimation is.

(4) *Q5: Approximate ELBO due to Stirling's approximation*. The reviewer is correct in pointing out that the ELBO claim would need to be relaxed due to the use of this approximation. However, we have found out that this term appears with opposite signs in $T_2$ and $T_4$ and thus cancels out. We will clarify this in the final version but thank the reviewer for the insightful comment.

(5) *Stochastic optimization may obfuscate results*. We first clarify that the CPU times and performances are directly comparable across all methods. Our results only include one source of stochasticity due to noisy gradient estimates arising from MC sampling. However, while we mention the possibility to use stochastic optimization techniques in lines 179–183, we refer to the use of a second source of stochasticity due to mini-batch optimization. None of our experiments actually exploit this. We will clarify this is the final version.

(6 & 7) *Minor edits*: Many thanks for your suggestions, we will include them in the final version and the supplement.

**R#3:** *(1) Standard VI*. We would like to highlight how, even though the variational inference scheme follows from standard arguments, by exploiting the structure of the model and the approximate posterior we increase the algorithm efficiency and avoid high-variance gradient estimates. Applying black-box variational inference naïvely to a stuctured posterior would require sampling $\mathbf{f}$, $\lambda^{\star}$, $M$ and $\{\mathbf{y}_m\}_{m=1}^{M}$ thus slowing down the algorithm while leading to poor convergence.

*(2) Q5: Clarify how superposition view helps*. The model double intractability arises first from the estimation of the integral of $\lambda(\mathbf{x})$ in Eq. (1) and second from the standard posterior estimation which requires the computation of the marginal likelihood. The augmentation scheme helps us with the first intractability. By superimposing two PPP with opposite intensities we obtain an homogeneous PPP and thus avoid the integration of the GP over $\mathcal{X}$. Instead, we only need to compute the measure of the input space $\int_{\mathcal{X}} d\mathbf{x}$, see Eq. (4). We will expand on this in the final version.

[Meta-Review · NeurIPS 2019]

A nice paper on a structured variational approximation for Cox processes with sigmoidal link functions. The key challenge is to represent the variational posterior over a set of thinned events with unknown cardinality. The key simplifying assumption is that the M thinned events are iid from a mixture of truncated normals; this seems like a strong assumption that should be discussed in greater detail in the main text. Please also take the reviewers' suggestions into account and follow through on the promises made in the rebuttal.